# Deep Reinforcement Learning for Computation Offloading and Resource Allocation in Unmanned-Aerial-Vehicle Assisted Edge Computing

**DOI:** 10.3390/s21196499

**Published:** 2021-09-29

**Authors:** Shuyang Li, Xiaohui Hu, Yongwen Du

**Affiliations:** School of Electronic and Information Engineering, Lanzhou Jiaotong University, Lanzhou 730070, China; 0619680@stu.lzjtu.edu.cn (S.L.); duyongwen@mail.lzjtu.cn (Y.D.)

**Keywords:** unmanned aerial vehicle, edge computing, computation offloading, resource allocation, soft actor–critic

## Abstract

Computation offloading technology extends cloud computing to the edge of the access network close to users, bringing many benefits to terminal devices with limited battery and computational resources. Nevertheless, the existing computation offloading approaches are challenging to apply to specific scenarios, such as the dense distribution of end-users and the sparse distribution of network infrastructure. The technological revolution in the unmanned aerial vehicle (UAV) and chip industry has granted UAVs more computing resources and promoted the emergence of UAV-assisted mobile edge computing (MEC) technology, which could be applied to those scenarios. However, in the MEC system with multiple users and multiple servers, making reasonable offloading decisions and allocating system resources is still a severe challenge. This paper studies the offloading decision and resource allocation problem in the UAV-assisted MEC environment with multiple users and servers. To ensure the quality of service for end-users, we set the weighted total cost of delay, energy consumption, and the size of discarded tasks as our optimization objective. We further formulate the joint optimization problem as a Markov decision process and apply the soft actor–critic (SAC) deep reinforcement learning algorithm to optimize the offloading policy. Numerical simulation results show that the offloading policy optimized by our proposed SAC-based dynamic computing offloading (SACDCO) algorithm effectively reduces the delay, energy consumption, and size of discarded tasks for the UAV-assisted MEC system. Compared with the fixed local-UAV scheme in the specific simulation setting, our proposed approach reduces system delay and energy consumption by approximately 50% and 200%, respectively.

## 1. Introduction

In the past decade, the exponential growth and diversity of Internet of Things (IoT) devices have changed the way we live [1]. Advances in 5G technology and the IoT have made many emerging applications possible, such as autonomous driving, smart cities, virtual reality (VR)/augmented reality (AR), real-time video analysis, and cloud gaming. Nevertheless, the traditional cloud-based computing paradigm is not suitable for those IoT terminal devices with limited computing and battery resources. The emergence of mobile edge computing (MEC) technology is expected to improve this situation. The MEC server deployed at the edge of the access network can provide terminal devices with computing and communication resources, bringing many benefits, such as reducing computing workload, delay, network congestion, and energy consumption. Traditional MEC servers are usually deployed on cellular base stations (BS) or Wi-Fi access points (AP). However, considering the construction cost and the gradual infrastructure update, not all BSs and APs can deploy edge servers. Therefore, mobile platforms, such as vehicles and UAVs are regarded as alternative candidates for MEC servers.

### 1.1. Motivation and Related Work

The existing literature has shown the great potential and capability of edge computing, which usually takes delay, energy consumption, system costs, etc., as the optimization objective [2,3,4,5,6,7,8,9,10]. However, all those studies assume that the stable wired or wireless communication link is established with sufficient bandwidth between the end-user and the distributed edge resources deployed fixedly. The existing computation offloading approaches are challenging to apply to specific scenarios, such as the dense distribution of end-users and the sparse distribution of network infrastructure.

With the advantages of flexible deployment, high mobility, strong line-of-sight (LOS) connection, and hover capability, UAVs are expected to play a critical role in wireless networks. Specifically, since UAVs can be deployed freely and flexibly in three-dimensional space, direct LOS communication with any terminal device can be easily established. This advantage allows UAVs to be used as relay nodes in wireless networks to improve communication between end-users. In addition, with the evolution of the UAV and chip industry, more computing and storage resources could be placed on UAVs, making it possible for UAVs to provide value-added computing services. UAVs not only play an essential role in the military domains but are also widely used in the civilian domains [11,12,13,14]. Due to its compelling features, UAVs as a kind of auxiliary computing and communication entity, which has been considered in the MEC technology. UAV-assisted edge computing technology could effectively provide computing and communication support for end-users in the special scenarios mentioned above. In UAV-assisted edge computing systems, the UAV can adjust its position in time according to the dynamic communication environment, ensuring that a reliable communication link can be established between the end-user and the MEC server. So far, tons of studies have shown the feasibility of UAV-assisted edge computing offloading technology.

However, in most previous studies [15,16], the computing capacity provided by UAVs was ignored in UAV-assisted wireless networks, and the communication capacity was mainly considered. Recent studies have begun to consider the computing resources of UAVs. Ref. [17,18,19,20] have only considered the communication and computing interaction between the two types of entities, which is the end-user offload part of the computing tasks to the UAV through wireless communication. Compared with UAVs, MEC servers have more computing and storage resources and are not limited by battery capacity. However, existing studies have rarely considered letting UAVs and MEC servers cooperate in providing end-user services. As far as we know, UAV-assisted MEC systems involving end-users, UAVs, and MEC servers have rarely been studied.

Reinforcement learning is a control-theoretic trial-and-error learning method [21]. The agent interacts with the environment and makes decisions through feedback from the environment. Existing studies have proved that reinforcement learning can handle offloading decision-making in the MEC environment well. Huang et al. [22] have proposed a DQN-based approach for computation offloading and resource allocation, which minimize the overall offloading cost of the MEC environment. Yang et al. [23] have proposed a DQN-based optimization approach for task scheduling in the UAV-assisted MEC environment, which could improve the efficiency of the task execution in each UAV. As far as we know, most DRL-based computation offloading approaches use the DQN algorithm. The selection of DRL algorithms depends highly on the dimension of state space and action space. Compared with the traditional Q-learning algorithm, the DQN algorithm applies the neural network to approximate the Q-value, which could handle high-dimensional state problems well. However, the DQN algorithm cannot handle high-dimensional action problems.

### 1.2. Main Contributions

To fill the previous studies, we envision a UAV-assisted MEC system consisting of multiple edge servers, multiple end-users, and a UAV. The UAV and MEC servers in the MEC system cooperatively work to provide computing services for end-users. With the help of the UAVs, the end-user can offload the tasks to the MEC server outside of its communication range and execute tasks by the MEC servers. We also consider that the computing task is partially offloaded, which is different from most existing studies’ binary offloading model and is closer to the actual situation [20]. We take the minimization of the weighted total cost of delay, energy consumption, and the size of discarded tasks as the optimization objective and further formulate the offloading decision problem as a Markov decision process. To this end, we propose a dynamic computation offloading approach based on the soft actor–critic (SAC) DRL algorithm. The SAC algorithm introduces entropy into the traditional actor–critic algorithm, improves the decision-making performance and obtains the global optimal policy. Numerical simulation results have proved the effectiveness of our proposed SAC-based dynamic computing offloading (SACDCO) algorithm compared with other baseline schemes. The differences between our work and the existing literature are summarized in Table 1.

The remainder of this paper is organized as follows. Section 2 describes the system model and problem formulation. Section 3 introduces the UAV-assisted edge computation offloading approach we proposed. The performance evaluation of our proposed approach is achieved through a series of simulations, and numerical results are given in Section 4. Section 5 summarizes the paper.

## 2. System Model and Problem Formulation

### 2.1. System Model

The UAV-assisted MEC system is composed of multiple edge servers, multiple end-users, and a UAV, which is shown in Figure 1. We consider a set of end-users, and each end-user periodically executes compute-intensive and delay-sensitive tasks during the decision episode. Due to signal congestion and the limited communication distance of the end-user, stable wireless communication cannot be established between the end-user and the MEC server. The UAV is equipped with antennas to communicate with end-users and MEC servers in the coverage area. In UAV-assisted MEC systems, end-users can offload computing tasks to UAVs. Compared with end-users, UAV has stronger computing power, but it still cannot compare with the computing power of MEC servers. Considering the limited battery capacity and computing power of the UAV, if the UAV cannot complete the computing task well, it will further consider offloading the computing task to the MEC server in the distance. The follow-me cloud (FMC) [27] controller is used in our proposed UAV-assisted MEC system, which could obtain the global information of end-users, MEC servers, and the UAV. Therefore, the proposed dynamic computing offloading algorithm is executed on the FMC controller.

Without loss of generality, a set of end-users is denoted by N={1,2,⋯,n,⋯,N}, a set of MEC servers is denoted by S={1,2,⋯,s,⋯,S}, and the UAV is denoted by U={u}. The entire decision episode is divided into multiple time slots, where T={1,2,⋯,t,⋯,T} denotes their corresponding set. The UAV stays at a fixed altitude $h_{u}\left(t\right)=H,\forall t\in T$. We define the three dimensional Cartesian coordinates of the UAV as $L_{u}\left(t\right)=\left[x_{u}\left(t\right),y_{u}\left(t\right),h_{u}\left(t\right)\right]$, and the coordinates of the end-user as $L_{n_{k}}=\left[x_{n_{k}},y_{n_{k}},0\right]$, and the coordinates of the MEC servers are $L_{s_{k}}=\left[x_{s_{k}},y_{s_{k}},0\right]$, where K∈(1,2,…,k,…,K) denotes the corresponding serial number. Unless otherwise stated, the important notations used in this paper are summarized in Table 2.

#### 2.1.1. Communication Model

The UAV offers services to all end-users but only serves one end-user in each time slot. We assume that all end-users are fixed at a certain coordinate $L_{n_{k}}=\left[x_{n_{k}},y_{n_{k}},0\right].$ At the beginning of the whole decision episode, the UAV *u* is deployed at the initial position $L_{u}\left(0\right)=\left[x_{u}\left(0\right),y_{u}\left(0\right),h_{u}\left(0\right)\right]$. When a certain end-user needs to provide services, the UAV flies directly above the end-user and establishes the communication link. Similar to [18], the communication links are presumed to be dominated by the LOS channels. Thus, the channel gain between end-user $n_{k}$ and the UAV *u* could be denoted as
(1)$$g_{n_{k},u}\left(t\right)=\alpha_{0}d_{n_{k},u}^{-2}\left(t\right)=\frac{\alpha_{0}}{{∥L_{n_{k}}-L_{u}\left(t\right)∥}^{2}}$$
where
$$∥L_{n_{k}}-L_{u}\left(t\right)∥=\sqrt{{\left[x_{n_{k}}-x_{u}\left(t\right)\right]}^{2}+{\left[y_{n_{k}}-y_{u}\left(t\right)\right]}^{2}+{\left[0-h_{u}\left(t\right)\right]}^{2}},$$
$d_{n_{k},u}\left(t\right)$ denotes the Euclidean distance between the end-user $n_{k}$ and the UAV *u*, ∥·∥ denotes the Euclidean norm, and $\alpha_{0}$ denotes the received power at the reference distance of 1 m for the transmission power of 1 W. Considering the blocking of the communication signal by the building, the wireless transmission rate can be denoted as
(2)$$r_{n_{k},u}\left(t\right)=B\log_{2}\left(1+\frac{P_{down}g_{n_{k},u}\left(t\right)}{\sigma^{2}+f_{n_{k},u}\left(t\right)P_{NLOS}}\right)$$
where *B* denotes the assigned communication bandwidth, $P_{down}$ denotes the received power of the UAV, $\sigma^{2}$ denotes the noise power, $P_{NLOS}$ denotes the transmission loss, $f_{n_{k},u}\left(t\right)$ denotes whether there is a communication block between end-user $n_{k}$ and the UAV in time slot *t* (that is, 0 means no blocking, and 1 means blocking). Similarly, when the UAV needs further to send the computing tasks to the remote MEC server, the channel gain between the UAV *u* and the MEC server $s_{k}$ could be denoted as
(3)$$g_{u,s_{k}}\left(t\right)=\alpha_{0}d_{u,s_{k}}^{-2}\left(t\right)=\frac{\alpha_{0}}{{∥L_{u}\left(t\right)-L_{s_{k}}∥}^{2}}$$
where
$$∥L_{u}\left(t\right)-L_{s_{k}}∥=\sqrt{{\left[x_{u}\left(t\right)-x_{s_{k}}\right]}^{2}+{\left[y_{u}\left(t\right)-y_{s_{k}}\right]}^{2}+{\left[h_{u}\left(t\right)-0\right]}^{2}},$$

Similarly, the wireless transmission rate between the UAV *u* and MEC server $s_{k}$ could be denoted as
(4)$$r_{u,s_{k}}\left(t\right)=B\log_{2}\left(1+\frac{P_{up}g_{u,s_{k}}\left(t\right)}{\sigma^{2}+f_{u,s_{k}}\left(t\right)P_{NLOS}}\right)$$

#### 2.1.2. Computation Model

Due to the limited computing resource of the end-user, our proposed offloading decision optimization algorithm is applied to each time slot. According to the offloading policy, the end-user offloads part of the tasks to the UAV, and then the UAV determines to process it locally or further offload to the MEC server. It should be noted that compared with the entire communication and calculation delay, the time to divide the task is very short, so this part of the delay is ignored in our model. In addition, in some computing-intensive applications, such as video analysis, the output data size of the computing results is often much smaller than the input data size. Therefore, the delay of the downlink is also ignored. The key components of the total delay during the offloading process are described as follows.

The flight delay from the previous location to the end-user directly above;The transmission delay from the end-user to the UAV;The calculation delay of the UAV;The calculation delay of the end-user;The transmission delay from the UAV to the MEC server;The calculation delay of the MEC server.

The flight delay from the previous location of the UAV *u* to the end-user directly above could be described as
(5)$$t_{fly}\left(t\right)=\frac{\sqrt{{\left[x_{n_{k}}\left(t\right)-x_{u}\left(t\right)\right]}^{2}+{\left[y_{n_{k}}\left(t\right)-y_{u}\left(t\right)\right]}^{2}}}{v_{u}}$$
where $v_{u}$ is the average flight speed of the UAV *u*. The transmission delay from end-user $n_{k}$ to UAV *u* could be described as
(6)$$t_{tr}^{u,n_{k}}\left(t\right)=\frac{R_{uav}\left(t\right)D_{n_{k}}\left(t\right)}{r_{u,n_{k}}\left(t\right)}$$
where $R_{uav}\left(t\right)\in \left[0,1\right]$ is the offloading rate of the end-user $n_{k}$ to the UAV, and $D_{n_{k}}\left(t\right)$ is the computing task size of the end-user $n_{k}$ in time slot *t*. The calculation delay of the UAV *u* could be described as
(7)$$t_{cal}^{u}\left(t\right)=\frac{R_{uav}\left(t\right)D_{n_{k}}\left(t\right)s}{f_{uav}}$$
where *s* denotes the CPU cycles required to process each byte, and $f_{uav}$ denotes the calculation frequency of the MEC servers’ CPU. Similar to (7), the local calculation delay of the end-user $n_{k}$ in time slot *t* could be denoted as
(8)$$t_{cal}^{n_{k}}\left(t\right)=\frac{\left(1-R_{uav}\left(t\right)\right)D_{n_{k}}\left(t\right)s}{f_{n_{k}}}$$
where $f_{n_{k}}$ denotes the calculation frequency of the end-user $n_{k}$. According to the offloading policy, it is decided whether to offload the computing tasks to the MEC servers. Due to the limited battery capacity, we consider offloading all the computing tasks received by the UAV to the MEC servers. Therefore, the transmission delay from the UAV *u* to the MEC server $s_{k}$ could be denoted as
(9)$$t_{tr}^{u,s_{k}}\left(t\right)=\frac{R_{uav}\left(t\right)D_{n_{k}}\left(t\right)}{r_{u,s_{k}}\left(t\right)}$$

The calculate delay of the MEC server $s_{k}$ could be denoted as
(10)$$t_{cal}^{s_{k}}\left(t\right)=\frac{R_{uav}\left(t\right)D_{n_{k}}\left(t\right)s}{f_{s_{k}}}$$

To define the service delay of each time slot, we assume that the UAVs and the MECs server can only start executing the computing tasks after the transmission is completed to ensure the reliability of the calculation result. We also assume that the end-users execute locally and transmit computing tasks at the same time. Based on the above assumption, the service delay of each time slot could be denoted as  
(11)$$T\left(t\right)=\left\{\begin{array}{cc} t_{cal}^{n_{k}}\left(t\right), & \mathrm{For}\;\mathrm{end}\text{-}\mathrm{user}\;\mathrm{only}.\\ t_{fly}\left(t\right)+t_{tr}^{u,n_{k}}\left(t\right)+t_{cal}^{u}\left(t\right), & \mathrm{For}\;\mathrm{end}\text{-}\mathrm{user}\;\mathrm{and}\;\mathrm{the}\;\mathrm{UAV}.\\ t_{fly}\left(t\right)+t_{tr}^{u,n_{k}}\left(t\right)+t_{tr}^{u,s_{k}}\left(t\right)+t_{cal}^{s_{k}}\left(t\right), & \mathrm{For}\;\mathrm{end}\text{-}\mathrm{user},\;\mathrm{the}\;\mathrm{UAV},\;\mathrm{and}\;\mathrm{the}\mathrm{MEC}\;\mathrm{server}. \end{array}\right.$$

#### 2.1.3. Energy Model

Battery capacity has always been a bottleneck in UAV applications. The battery capacity of the UAV is denoted as $E_{\mathrm{battery}\;}$. At the beginning of the decision episode, the UAV is in a fully charged state. The UAV continues to serve the end-user until the battery capacity is exhausted. Our study mainly focuses on the calculation and transmission energy consumption of the UAV while ignoring other energy consumption, which has nothing to do with our decision-making. The key components of the energy consumption during the offloading process are described as follows.

The flight energy consumption of the UAV;The transmission energy consumption when UAV receives tasks from end-users;The calculation energy consumption of the UAV;The transmission energy consumption from the UAV to the MEC server.

The flight energy consumption of the UAV could be denoted as
(12)$$E_{fly}\left(t\right)=Pt_{fly}\left(t\right)=Fv_{u}t_{fly}\left(t\right)$$
where $F=m_{uav}*g$, which is related to the weight of the UAV. The transmission energy consumption when UAV receives tasks from end-users could be denoted as
(13)$$E_{tr}^{n_{k},u}\left(t\right)=P_{down}t_{tr}^{u,n_{k}}\left(t\right)$$
where $P_{down}$ denotes the received power of the UAV, and $t_{tr}^{u,n_{k}}\left(t\right)$ denotes the transmission delay. Similar to [28], we model that the calculation power is positively correlated with computing capacity, i.e., $\kappa {\left(f_{uav}\right)}^{3}$, where κ denotes the energy consumption factor. The UAV calculation energy consumption is denoted as
(14)$$E_{cal}^{u}\left(t\right)=\kappa {\left(f_{uav}\right)}^{3}t_{cal}^{u}\left(t\right)$$

The sending power of the UAV is denoted as $P_{up}$, and the transmission energy consumption of the UAV could be denoted as
(15)$$E_{tr}^{u,s_{k}}\left(t\right)=P_{up}t_{tr}^{u,s_{k}}\left(t\right)$$

According to the above analysis, the total energy consumption of the UAV could be denoted as
(16)$$E_{u}\left(t\right)=\left\{\begin{array}{cc} E_{fly}\left(t\right)+E_{tr}^{n_{k},u}\left(t\right)+E_{cal}^{u}\left(t\right), & \mathrm{For}\;\mathrm{UAV}\;\mathrm{and}\;\mathrm{end}\text{-}\mathrm{user}.\\ E_{fly}\left(t\right)+E_{tr}^{n_{k},u}\left(t\right)+E_{tr}^{u,n_{k}}\left(t\right), & \mathrm{For}\;\mathrm{UAV},\;\mathrm{end}\text{-}\mathrm{user},\;\mathrm{and}\;\mathrm{MEC}\;\mathrm{server}. \end{array}\right.$$

### 2.2. Problem Formulation

Our study objective is to minimize the weighted total cost of the service delay, energy consumption of the UAV, and the size of the discarded tasks through optimize the offloading policy. The joint optimization problem could be denoted as
(17)$$\min_{L_{u}\left(t\right),R_{uav}\left(t\right),R_{mec}\left(t\right)}\sum_{t\in T}\left\{E_{u}\left(t\right)+\rho_{1}T\left(t\right)+\rho_{2}S\left(t\right)\right\}$$
(18)$$s.t.\;L_{u}\left(t\right)\in \{\left(x\left(t\right),y\left(t\right)|x\left(t\right)\in \left[0,L\right],y\left(t\right)\in \left[0,W\right]\right\},\forall \;t\in T$$
(19)$$\sum_{t=1}^{T}E_{u}\left(t\right)\;\leq E_{battery},\forall \;t\in T$$
(20)$$0<R_{uav}\left(t\right)<1,\forall \;t\in T$$
(21)$$R_{mec}\left(t\right)\in \left\{0,1\right\},\forall \;t\in T$$
(22)$$\sum_{t=1}^{T}\sum_{k=1}^{K}D_{n_{k}}\left(t\right)=D,\forall \;t\in T,\forall \;k\in K$$
(23)$$0<S\left(t\right)<D_{n_{k}}\left(t\right),\forall \;t\in T$$
where $\rho_{1},\rho_{2}>0$ in (Equation 17) are the parameters that define the relative weight, and S(t) denotes the total size of the discarded tasks in time slot *t*. Constraint (Equation 18) limits the UAV’s range of movement. Constraint (Equation 19) means the total energy consumption during the decision episode cannot exceed the maximum battery capacity of the UAV. Constraint (Equation 20) denotes the value range of the offloading ratio. Constraint (Equation 21) denotes whether to further offload to the MEC server. In (Equation 22), *D* denotes the total size of computing tasks that should be executed during the decision episode. Constraint (Equation 23) denotes that the size of the discarded tasks does not exceed the total size of computing tasks in each time slot.

## 3. Soft Actor–Critic Based Dynamic Computation Offloading Algorithm

Our study objective is to obtain the optimal offloading policy by minimizing the weighted total cost of the service delay, energy consumption of the UAV, and the size of the discarded task during the entire decision episode. We consider the standard reinforcement learning framework [29] and formulate the UAV-assisted edge computing offloading decision-making and resource-allocating problem as a Markov decision process (MDP).

### 3.1. Markov Decision Process

The MDP is usually described as a quintuple M=〈S,A,P,R,γ〉, which denotes state, action, state transition probability, reward, and discount factor, respectively. The FMC controller used in our proposal can obtain all global information of the end-user, the UAV, and the MEC server. Therefore, the DRL-based algorithm we proposed runs on the FMC controller in each time slot. Furthermore, the state space, action space, and reward are defined as follows.

State space: We consider the current location of the UAV $L_{u}\left(t\right)$, the UAV battery capacity $E_{\mathrm{battery}\;}\left(t\right)$, and the size of computing tasks $D_{n_{k}}\left(t\right)$ as the current state. Therefore, the state space can be denoted as
(24)$$s\left(t\right)=\left\{L_{u}\left(t\right),E_{battery}\left(t\right),D_{n_{k}}\left(t\right)\right\}$$Action space: We consider the offloading rate $R_{uav}\left(t\right)$, whether to further offload to the MEC servers $R_{s_{k}}\left(t\right)$ as the current action of the agent. Therefore, action space can be denoted as
(25)$$a\left(t\right)=\left\{R_{uav}\left(t\right),R_{s_{k}}\left(t\right)\right\}$$Reward: We define cumulative rewards to minimize the weighted sum of service delay, energy consumption, and the size of discarded task. Thus, rewards can be denoted as
(26)$$r\left(t\right)=-\left(E_{u}\left(t\right)+\rho_{1}*T\left(t\right)+\rho_{2}*S\left(t\right)\right)$$
where $\rho_{1},\rho_{2}>0$ denote the relative weight.

### 3.2. Soft Actor–Critic DRL Algorithm

Previous studies have shown that DRL algorithms can solve the offloading decision problems in the MEC environment [21,22,30]. However, those DRL algorithms suffer from two main problems: high sample complexity (large amounts of data needed) and other being their brittleness with respect to learning rates, exploration constants, and other hyperparameters. Algorithms, such as DDPG and twin delayed DDPG (TD3), are used to tackle the challenge of high sample complexity in actor–critic frameworks with continuous action spaces. However, they still suffer from brittle stability with respect to their hyperparameters.

Soft actor–critic (SAC) algorithm introduces an actor–critic framework for arrangements with continuous action spaces where in the standard objective of reinforcement learning, i.e., maximizing expected cumulative reward, is augmented with an additional objective of entropy maximization, which provides a substantial improvement in exploration and robustness. Thus, the optimization objective of SAC algorithm is described as
(27)$$\pi^{*}=\arg\max_{\pi }\sum_{t}E_{\left(s_{t},a_{t}\right)∼\rho_{\pi }}\left[r\left(s_{t},a_{t}\right)+\alpha H\left(\pi \left(\cdot |s_{t}\right)\right)\right]$$
where α>0 is the temperature parameter, which determines the relative importance of the entropy term against the reward. H represents the entropy function. The entropy of a random variable *x* following a probability distribution *P* is defined as $H\left(P\right)=E_{x∼P}\left[-\log P\left(x\right)\right]$. Similar to the traditional actor–critic algorithm, value function $V^{\pi }\left(s\right)$ and state-value function $Q^{\pi }\left(s,a\right)$ could be defined in the SAC algorithm, which are given as follows
(28)$$V^{\pi }\left(s\right)=\underset{\tau ∼\pi }{E}\left[\sum_{t=0}^{\infty }\gamma^{t}\left(R\left(s_{t},a_{t},s_{t+1}\right)+\alpha H\left(\pi \left(\cdot |s_{t}\right)\right)\right)|s_{0}=s\right]$$
(29)$$Q^{\pi }\left(s,a\right)=\underset{\tau ∼\pi }{E}\left[\sum_{t=0}^{\infty }\gamma^{t}R\left(s_{t},a_{t},s_{t+1}\right)+\alpha \sum_{t=1}^{\infty }\gamma^{t}H\left(\pi \left(\cdot |s_{t}\right)\right)|s_{0}=s,a_{0}=a\right]$$

According to above analysis, $V^{\pi }$ and $Q^{\pi }$ are connected by:(30)$$V^{\pi }\left(s\right)=\underset{a∼\pi }{E}\left[Q^{\pi }\left(s,a\right)\right]+\alpha H\left(\pi \left(\cdot |s\right)\right)$$
and the Bellman equation for $Q^{\pi }$ is
(31)Qπ(s,a)=Es′∼Pa′∼πRs,a,s′+γQπs′,a′+αHπ·∣s′=Es′∼PRs,a,s′+γVπs′

SAC learns a policy $\pi_{\theta }$ and two *Q* functions $Q_{\phi_{1}},Q_{\phi_{2}}$ and their target networks concurrently. The two Q-functions are learned in a fashion similar to TD3, where a common target is considered for both the Q functions, and clipped double Q-learning is used to train the network. The action-value for one state-action pair can be approximated as
(32)$$Q^{\pi }\left(s,a\right)\approx r+\gamma \left(Q^{\pi }\left(s^{\prime },{\tilde{a}}^{\prime }\right)-\alpha \log\pi \left({\tilde{a}}^{\prime }|s^{\prime }\right)\right),\;{\tilde{a}}^{\prime }∼\pi \left(\cdot |s^{\prime }\right)$$
where ${\tilde{a}}^{\prime }$ (action taken in next state) is sampled from the policy.

SAC also uses replay buffer like other off-policy algorithms. The quintuple $\left(s,a,r,s^{\prime },d\right)$ from each episode is stored into the replay buffer D. Batches of these transitions are sampled while updating the network parameters.

Just like TD3, SAC uses clipped double Q-learning to calculate the target values for the Q-value network. The target is given by
(33)$$y^{t}\left(r,s^{\prime },d\right)=r+\gamma \left(\min_{j=1,2}Q_{\phi_{\mathrm{targ}\;},}\left(s^{\prime },{\tilde{a}}^{\prime }\right)-\alpha \log\pi_{\theta }\left({\tilde{a}}^{\prime }|s^{\prime }\right)\right)$$
where ${\tilde{a}}^{\prime }$ is sampled from the policy. The loss function can be defined as
(34)$$L\left(\phi_{i},D\right)=E_{\left(s,a,r,s^{\prime },d\right)∼D}\left[{\left(Q_{\phi_{i}}\left(s,a\right)-y^{t}\left(r,s^{\prime },d\right)\right)}^{2}\right]$$

The main objective of policy optimization will be to maximize the value function, which, in this case, can be defined as
(35)$$V^{\pi }\left(s\right)=E_{a∼\pi }\left[Q^{\pi }\left(s,a\right)-\log\pi \left(a|s\right)\right]$$

In SAC, a reparameterization trick is used to sample actions from the policy to ensure that sampling from the policy is a differentiable process. The policy is now parameterized as
(36)$${\tilde{a}}_{t}^{\prime }=f_{\theta }\left(\xi_{t};s_{t}\right)$$
(37)$${\tilde{a}}_{\theta }^{\prime }\left(s,\xi \right)=\tanh\left(\mu_{\theta }\left(s\right)+\sigma_{\theta }\left(s\right)⊙\xi \right)$$
(38)ξ∼N(0,1)

The maximization objective is now defined as
(39)$$\max_{\theta }E_{\left(s∼D,\xi ∼N\right)}\left[\min_{j=1,2}Q_{\phi_{j}}\left(s,{\tilde{a}}_{\theta }\left(s,\xi \right)\right)-\alpha \log\pi_{\theta }\left({\tilde{a}}_{\theta }\left(s,\xi \right)|s\right)\right]$$

The pseudocode of soft actor–critic algorithm is given in Algorithm 1 [31]. We optimize the computation offloading policy via the soft actor–critic DRL algorithm in each time slot, thereby minimize the optimization objective. The pseudocode of the SAC-based UAV-assisted computation offloading algorithm is given in Algorithm 2.
**Algorithm 1** Soft actor–critic algorithm**Input: **$\theta_{1},\theta_{2},\phi$  1:${\bar{\theta }}_{1}\leftarrow \theta_{1},{\bar{\theta }}_{2}\leftarrow \theta_{2}$  2:D←∅  3:**for** each iteration **do**  4:    **for** each environment step **do**  5:        $a_{t}∼\pi_{\phi }\left(a_{t}|s_{t}\right)$  6:        $s_{t+1}∼p\left(s_{t+1}|s_{t},a_{t}\right)$  7:        $D\leftarrow D\cup \left\{\left(s_{t},a_{t},r\left(s_{t},a_{t}\right),s_{t+1}\right)\right\}$  8:    **end for**  9:    **for** each gradient step **do**10:        $\theta_{i}\leftarrow \theta_{i}-\lambda_{Q}{\hat{\nabla }}_{\theta_{i}}J_{Q}\left(\theta_{i}\right)$ for i∈{1,2}11:        $\phi \leftarrow \phi -\lambda_{\pi }{\hat{\nabla }}_{\phi }J_{\pi }\left(\phi \right)$12:        $\alpha \leftarrow \alpha -\lambda {\hat{\nabla }}_{\alpha }J\left(\alpha \right)$13:        ${\bar{\theta }}_{i}\leftarrow \tau \theta_{i}+\left(1-\tau \right){\bar{\theta }}_{i}$ for i∈{1,2}14:    **end for**15:**end for****Output: **$\theta_{1},\theta_{2},\phi$


**Algorithm 2** SAC-based dynamic computation offloading algorithm (SACDCO)**Input:** The initial location of the UAV $L_{u}\left(t\right)$, the initial battery capacity of the UAV $E_{\mathrm{battery}\;}\left(t\right)$, and task size $D_{n_{k}}(t).$
  1:**for** each time slot t∈T **do**  2:    Observe state $s\left(t\right)=\left\{L_{u}\left(t\right),E_{\mathrm{battery}\;}\left(t\right),D_{n_{k}}\left(t\right)\right\}$ and select action $a\left(t\right)=\left\{R_{uav}\left(t\right),R_{s_{k}}\left(t\right)\right\}$ based on policy parameter θ;  3:    Select an end-user $n_{k}$ and generate signal block randomly;  4:    **if**
$R_{uav}\left(t\right)!=0$
**then**  5:        The UAV flies directly above the end-user $n_{k}$;  6:        Calculate the flight distance of the UAV via (Equation 5);  7:        **if** flight distance !=0 **then**  8:            Calculate flight delay and flight energy consumption via (Equation 5) and (Equation 11);  9:        **end if** 10:       **if**
$R_{s_{k}}\left(t\right)==1$
**then** 11:            Calculate the channel gain $g_{n_{k},u}\left(t\right)$ and $g_{u,s_{k}}\left(t\right)$ via (Equation 1) and (Equation 3); 12:            Calculate transmission rate $r_{n_{k},u}\left(t\right)$ and $r_{u,s_{k}}\left(t\right)$ via (Equation 2) and (Equation 4); 13:            Calculate the transmission delay $t_{tr}^{u,n_{k}}\left(t\right)$ via (Equation 6), the transmission delay $t_{tr}^{u,s_{k}}\left(t\right)$ (Equation 9), and the calculation delay $t_{cal}^{s_{k}}\left(t\right)$ via (Equation 10), respectively; 14:            Calculate actual delay via (Equation 16); 15:            Calculate energy consumption via (Equation 15); 16:        **else** 17:            Calculate transmission delay, calculation delay of the UAV and via (Equation 6)–(Equation 8), respectively; 18:            Calculate actual delay via (Equation 16); 19:            Calculate UAV’s energy consumption of transmission and calculation via (Equation 12) and (Equation 13); 20:            Calculate actual energy consumption via (Equation 15); 21:        **end if** 22:    **else** 23:        Calculate local calculation delay via (Equation 8); 24:    **end if** 25:    Calculate episode reward via (Equation 26); 26:    Update policy parameter θ via Algorithm 1. 27:**end for**



## 4. Performance Evaluation

In this section, a detailed numerical evaluation is presented to study the performance of our proposed SACDCO algorithm compared to other baseline schemes. In our proposal, all algorithms and the corresponding simulations are implemented based on Python and executed on a desktop computer with Intel Core i7-8700 6 cores CPU and 32 GB RAM.

### 4.1. Simulation Settings

As mentioned above, our proposed UAV-assisted MEC system consists of three entities: end-users, UAVs, and edge servers, which is different from previous studies. The offloading approaches in existing literature are not directly applicable to our settings. Thereby, we consider the following intuitive schemes as the baseline schemes.

(*Local-Only Scheme*) Only execute computing tasks by the end-user;(*UAV-Only Scheme*) Only execute computing tasks on the UAV without further offloading to any MEC servers;(*Fixed Local-UAV Scheme*) Half of the computing tasks is executed locally while the other half is executed on the UAV.

In our proposal, we consider a two-dimensional square area, in which four end-users are distributed in an area of $L\times W=500\times 500\;m^{2}$ and fixed positions. Furthermore, four MEC servers are deployed at the edge of the area, and each MEC server is equipped with 8 cores 3.0 GHz CPU. At the beginning of the decision episode, we assume that the UAV is deployed at an initial position $L_{u}=\left(250,250\right)$ with a height of *H* = 100 m. For UAV, we refer to the parameters of DJI Air 2S [32]. Unless otherwise stated, the simulation parameters are summarized in Table 3.

### 4.2. Simulation Result

In this section, we have verified the convergence of the SACDCO algorithm and the simulation results of different hyperparameters on the SACDCO algorithm to select the optimal hyperparameters. To prove the importance of offloading policy optimization, we have compared our proposed SACDCO approach with other baseline schemes regarding the delay, energy consumption, and the size of discarded tasks. Subsequently, we have also studied the influence of different UAV parameters on offloading decision-making. The specific simulation results and corresponding charts are given as follows.

Firstly, we study the impact of different learning rates on the cumulative reward of the SACDCO algorithm. As shown in Figure 2, when the learning rate is set to 0.003 and 0.0003, higher cumulative rewards can be obtained. Compared with the learning rate of 0.003, when the learning rate is 0.0003, the cumulative reward of the SACDCO algorithm can converge faster. Therefore, the following experimental results will be based on the learning rate set to 0.0003.

Secondly, we study the impact of different relative weight $\rho_{1}$, $\rho_{2}$ on the cumulative reward of the SACDCO algorithm. Considering the value range of the delay, energy consumption, and the size of discarded tasks, we design four schemes and conduct the corresponding simulation. According to the simulation result shown in Figure 3, we find that the SACDCO algorithm could get the highest cumulative reward when $\rho_{1}$ set to 0.01, and $\rho_{2}$ set to 0.1. Similarly, the following experimental results will be based on $\rho_{1}$ set to 0.01 and $\rho_{2}$ set to 0.1.

To prove the importance of offloading policy optimization, we compared our proposal with other baseline schemes in terms of delay, as shown in Figure 4. Since it is not affected by the local calculation delay, the UAV-only scheme has the lowest delay, about 32 s. After the SACDCO algorithm optimizes the offloading policy, the delay converges to about 40 s. The delays of the other two schemes are relatively high, around 80 s and 90 s, respectively.

The comparison of different schemes in terms of energy consumption of UAV is shown in Figure 5. Since the local-only scheme does not consume the energy of the UAV, the corresponding result does not appear in the figure. We find that the energy consumption generated by our proposed SACDCO approach is at a low level, which is about 2900 KJ (accounted for 60% of the highest energy consumption scheme only).

As shown in Figure 6, we compare our proposal with UAV-only and fixed local-UAV in terms of the size of discarded tasks. When UAV’s battery runs out, the remainder of offloaded tasks on the UAV will be discarded. The simulation result shows that the size of the discarded tasks generated by our proposed SACDCO algorithm is at a low level, which is about 35 Mbit (accounted for 43% of the size of total tasks).

We then study the impact of UAV computing capability and bandwidth on the weighted sum of delay, energy consumption, and the size of discarded tasks. We adjust the UAV’s computing power by changing the number of UAV’s CPU cores, and the simulation result is shown in Figure 7. The corresponding result shows that the proposed SACDCO algorithm could obtain the highest cumulative rewards when the UAV is equipped with two cores. Then, we study how cumulative rewards behave as the UAV bandwidth increase from 5GHz to 30GHz. We observe that as the bandwidth increases, the cumulative reward of the SACDCO algorithm will also increase. Compared with other baseline algorithms, the SACDCO algorithm can maintain good performance. It should be noted that too high bandwidth is usually impractical. The simulation result is given in Figure 8.

## 5. Conclusions

Considering stochastic computation tasks generated by end-users, the mobility of the UAV, and the limited battery capacity of the UAV, we have studied the computation offloading decision problem in the UAV-assisted MEC environment with multiple users and multiple MEC servers. To obtain the global optimal offloading policy, we minimize the weighted total cost of system delay, energy consumption, and the size of discarded tasks as the optimization objective. We propose the soft actor–critic dynamic computation offloading approach to optimize computation offloading and resource allocation policy. Unlike previous studies, we consider letting UAV and MEC servers work collaboratively to provide computing services for end-users. To this end, we consider three intuitive schemes as the baseline schemes in the simulation, i.e., local-only scheme, UAV-only scheme, and fixed local-UAV scheme, respectively. Extensive simulations have demonstrated the superiority of our proposal in terms of delay, energy consumption, and the size of discarded tasks. In particular, compared with the fixed local-UAV scheme in the specific simulation setting, our proposed approach reduces system delay and energy consumption by approximately 50% and 200%, respectively.

In the future, we will extend the research on offloading decision optimization to a multi-UAV collaborative edge computing scenario. Existing literature has shown the superiority of multi-UAV collaborative assisted edge computing. In [23], multiple UAVs work collaboratively to service IoT devices, and a DQN-based optimization approach was proposed to improve the efficiency of each UAV while maintaining the QoS of ground IoT devices. Nevertheless, the DQN algorithm could not handle high-dimensional action problems well, as we mentioned before. Savkin et al. [33] proposed a multi-UAV collaborative approach for improving the network performance between the UAV and the base stations. Chen [34] proposed an approach for improving pairing rate through optimizing the power allocations, UAVs’ locations, and nodes scheduling. However, their optimization objective only focuses on system delay, which is different from ours. In the future, we will introduce multi-agent reinforcement learning algorithms to our proposal, which could further improve the algorithm’s performance. We hold the opinion that multi-agent reinforcement learning algorithms will play an indispensable role in complex decision problems.

## Figures and Tables

**Figure 1 sensors-21-06499-f001:**
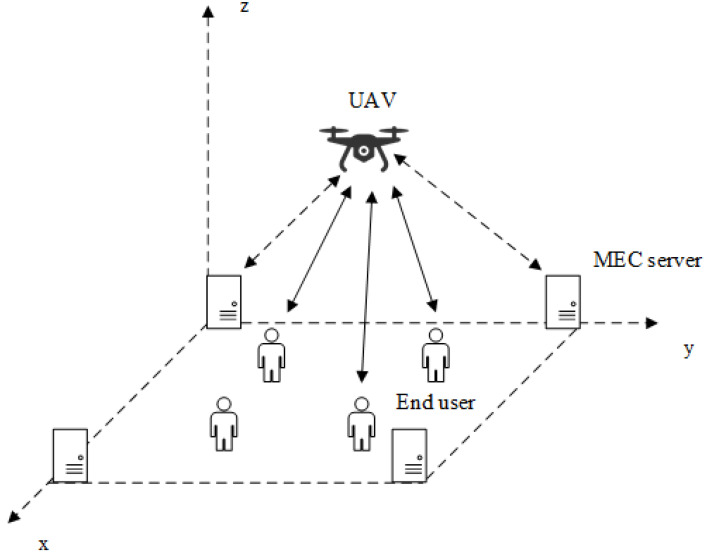
The architecture of UAV-assisted mobile edge computing system.

**Figure 2 sensors-21-06499-f002:**
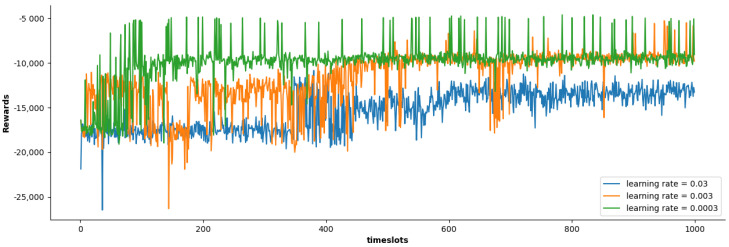
Comparison of cumulative rewards of SACDCO algorithm under different learning rates.

**Figure 3 sensors-21-06499-f003:**
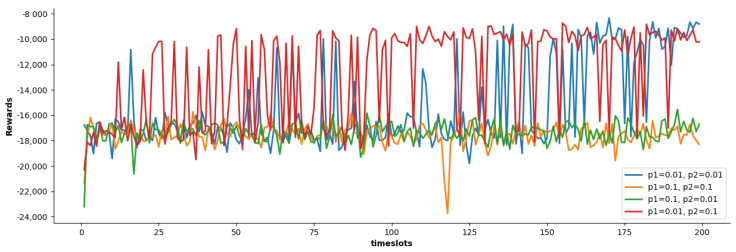
Comparison of cumulative rewards of SACDCO algorithm under different relative weight.

**Figure 4 sensors-21-06499-f004:**
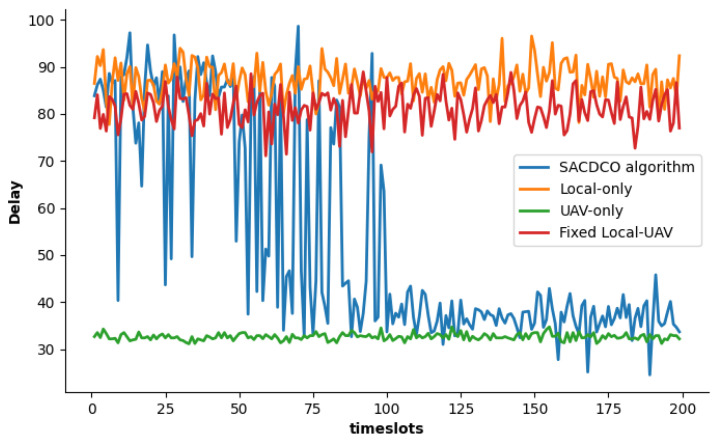
Comparison of delay under different schemes.

**Figure 5 sensors-21-06499-f005:**
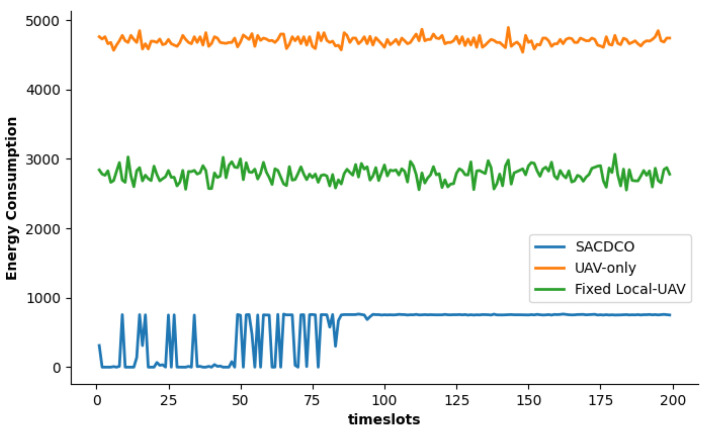
Comparison of energy consumption under different schemes.

**Figure 6 sensors-21-06499-f006:**
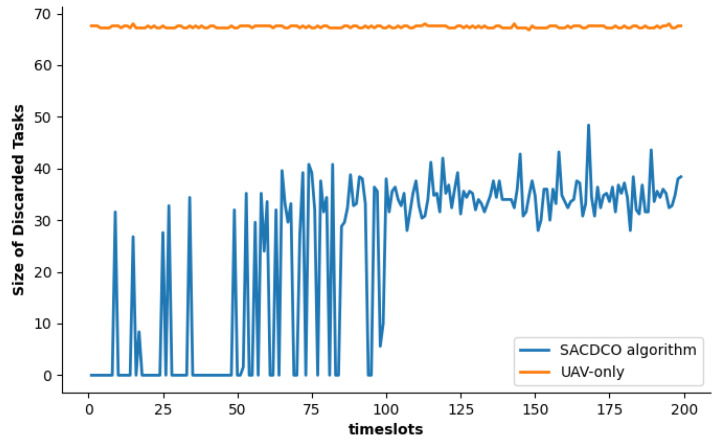
Comparison of the size of discarded tasks under different schemes.

**Figure 7 sensors-21-06499-f007:**
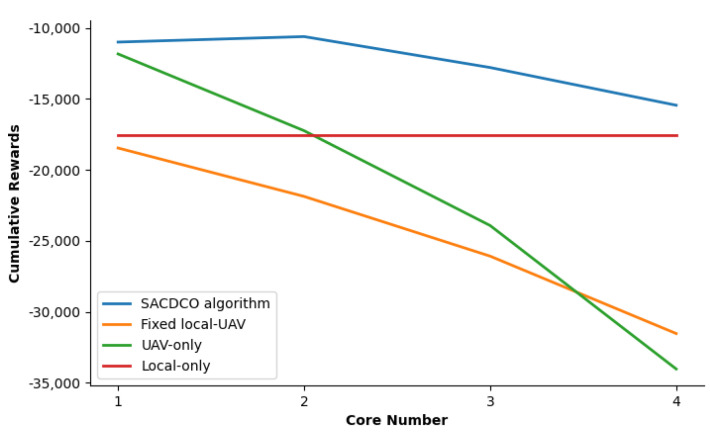
Comparison of cumulative rewards for different UAV computing capabilities under different schemes.

**Figure 8 sensors-21-06499-f008:**
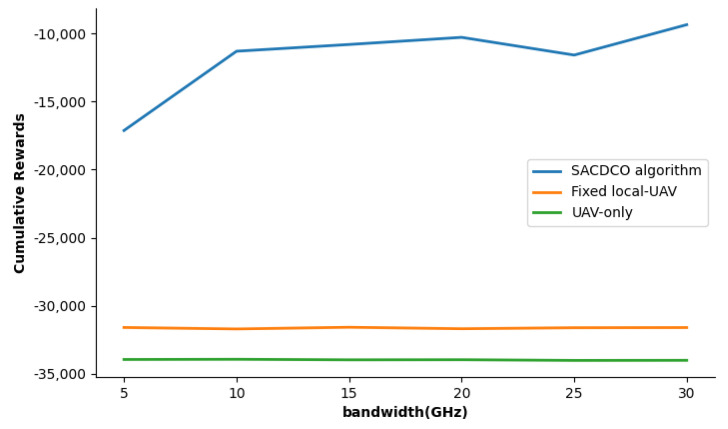
Comparison of cumulative rewards for different UAV bandwidth under different schemes.

**Table 1 sensors-21-06499-t001:** Comparison between our work and the existing literature. (✓) indicates that the topic is covered.

Reference	Communication-Only	Communication and Computation	EU-UAV	EU-UAV-MEC	Partial Offloading	RL Algorithm	Optimization Objective
[15]	✓						Throughput
[16]	✓						Throughput
[17]		✓	✓		✓		Delay
[18]		✓	✓				Energy consumption
[19]		✓	✓				Computation rate
[20]		✓	✓				Energy consumption
[22]		✓	✓			DQN	The cost of energy, computation, and delay
[23]		✓	✓			DQN	Load balance
[24]		✓	✓		✓	DQN	System utility
[25]		✓	✓		✓	Actor–Critic	Average response time
[26]		✓		✓		DQN	System reward
Our work		✓		✓	✓	Soft Actor–Critic	The cost of delay, energy, and discarded tasks

**Table 2 sensors-21-06499-t002:** List of Notations.

Notations	Definitions
N	The set of end-user *n*
S	The set of MEC server *s*
U	The set of unmanned aerial vehicle *u*
T	The set of time slot *t*
K	The maximum number of end users or MEC servers
$L_{n_{k}}\left(t\right)$	The location of the end-user $n_{k}$
$L_{s_{k}}\left(t\right)$	The location of the MEC server $s_{k}$
$L_{u}\left(t\right)$	The location of the UAV
$g_{n_{k},u}\left(t\right)$	The channel gain between the end-user $n_{k}$ and the UAV *u*
$r_{n_{k},u}\left(t\right)$	The transmission rate between the end-user $n_{k}$ and the UAV *u*
$g_{u,s_{k}}\left(t\right)$	The channel gain between the UAV *u* and the MEC server $s_{k}$
$r_{u,s_{k}}\left(t\right)$	The transmission rate between the UAV *u* and the MEC server $s_{k}$
$t_{fly}\left(t\right)$	The flight delay of UAV *u*
$t_{tr}^{u,n_{k}}\left(t\right)$	The transmission delay between the end-user $n_{k}$ and the UAV
$t_{cal}^{u}\left(t\right)$	The channel gain between the MEC server $s_{k}$ and the UAV *u*
$t_{cal}^{n_{k}}\left(t\right)$	The calculation delay of the end-user $n_{k}$
$t_{tr}^{u,s}\left(t\right)$	The transmission delay between the UAV and the MEC server $s_{k}$
$D_{n_{k}}\left(t\right)$	The computing tasks that end-user $n_{k}$ needs to complete
$R_{uav}\left(t\right)$	The offloading ratio of UAV
$R_{s_{k}}\left(t\right)$	Whether to further offload to the MEC server $s_{k}$
S(t)	The total size of the discarded tasks in time slot *t*

**Table 3 sensors-21-06499-t003:** Simulation Parameters.

Parameters	Values	Parameters	Values
$\alpha_{0}$	−50 dBm	*L*	500 m
$B_{n_{k}}$	10 MHz	*W*	500 m
$B_{uav}$	30 MHz	*H*	100 m
$P_{up}$	0.1 w	$L_{n_{1}}$	(125,125)
$P_{down}$	1 w	$L_{n_{2}}$	(375,125)
$\sigma^{2}$	−100 dBm	$L_{n_{3}}$	(125,375)
$P_{NLOS}$	−80 dBm	$L_{n_{4}}$	(375,375)
$v_{u}$	15 m/s	$L_{s_{1}}$	(0,0)
$D_{n_{k}}$	80 Mbit	$L_{s_{2}}$	(500,0)
*s*	1000	$L_{s_{3}}$	(0,500)
$m_{uav}$	0.6 kg	$L_{s_{4}}$	(500,500)
$E_{battery}$	$1.5\times 10^{5}$ J	$f_{n_{1}}$	0.4 GHz
$f_{uav}$	3.0 GHz × 2	$f_{n_{2}}$	0.6 GHz
$f_{mec}$	3.0 GHz × 4	$f_{n_{3}}$	0.8 GHz
*g*	10 m/s	$f_{n_{4}}$	1.0 GHz
*K*	$1\times 10^{-28}$		

## Data Availability

Not applicable.

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
