# Peer review of "Deep Reinforcement Learning for Computation Offloading and Resource Allocation in Unmanned-Aerial-Vehicle Assisted Edge Computing"

_sensors, 2021, doi:10.3390/s21196499_

Round 1
Reviewer 1 Report
More conclusions are nedeed about the analysed characteristics
Author Response
Point 1: More conclusions are needed about the analysed characteristics.
Response 1: We have updated the conclusion of the manuscript, highlighting it in yellow. The updated content includes some simulation results and future work, which are more detailed than the original manuscript.

Reviewer 2 Report
The paper analyzes the usage of deep learning in Unmanned-Aerial-Vehicle applications for optimal resources allocation. This research is in the scope of the journal and presents an efficient methodology based on UAVs to deal with dense distribution end-users. Authors claim that their approach is more efficient than existing literature methodologies, which is proved in the Results section.
My comments to improve the quality of this contribution are the following:
- In the introduction section, will be interesting to include a table that summarizes the main literature approaches in order to identify in an easy manner the gap in the literature that this research fills.
- I recommend rearranging the introduction section with the following structure. 1.1. General context, 1.2. Motivation, 1.3. Literature review, 1.4. contribution and scope., and 1.5. Document organization.
- The list of notations does not fulfill MDPI requirements, it must appear at the end of the document as a nomenclature.
- Numerical results are well structure and presented. My recommendation is to increase the size of the axis in some figures to allows easy visualization of the results.
- In the introduction please include some numerical achievements of this research.
Good job.
Author Response
Point 1: In the introduction section, will be interesting to include a table that summarizes the main literature approaches in order to identify in an easy manner the gap in the literature that this research fills.
Response 1: We have updated the introduction part of the manuscript based on your recommendation. The existing relevant literature is summarized in Table 1, which clearly shows the research gaps that our work fills.
Point 2: I recommend rearranging the introduction section with the following structure. 1.1. General context, 1.2. Motivation, 1.3. Literature review, 1.4. contribution and scope., and 1.5. Document organization.
Response 2: Thank you for your recommendation. We have rearranged the introduction part of the manuscript based on your recommendation, but the subtitle is slightly different. The updated content is highlighted in yellow in the attached manuscript file.
Point 3: The list of notations does not fulfill MDPI requirements, it must appear at the end of the document as a nomenclature.
Response 3: The LaTeX template we used is provided by the MDPI website, but we do not find any requirements related to the list of notations. We have also referred to other literature from MDPI, and they have not put the list of notations on the end of the document as a nomenclature. We are puzzled.
Point 4: Numerical results are well structure and presented. My recommendation is to increase the size of the axis in some figures to allows easy visualization of the results.
Response 4: Thank you for your recommendation. We have realized the problem and have updated the related figures, which include Figure 2 and Figure 3.
Point 5: In the introduction, please include some numerical achievements of this research.
Response 5: We have updated the numerical results of our proposal in the abstract and conclusion part of the manuscript, which is highlighted in yellow in the attached manuscript file.

Reviewer 3 Report
Although I understand and appreciate the research approach and methodology of this paper, I think the topic, the simulation and results should be better explained and positioned within a more realistic scenario.
I recommend you to dedicate a section or paragraph to give some examples and describe some real-life use-cases where the obtained results could o would create a difference.
Where do you position SACDCO compared to the Fixed Local-UAV Scheme in terms of percentage of allocation of tasks for UAV's ?
In UAV-Only Scheme and Fixed Local-UAV Scheme, having in mind that discarded tasks are correlated with UAV’s battery , why you don't implement a very simple algorithm to reduce distributions of tasks for UAV's that are approaching to the end of battery state ? Could you consider such an optimization in order to reduce the discarded tasks values ?
Author Response
Point 1: Although I understand and appreciate the research approach and methodology of this paper, I think the topic, the simulation and results should be better explained and positioned within a more realistic scenario. I recommend you dedicate a section or paragraph to give some examples and describe some real-life use-cases where the obtained results could create a difference.
Response 1: We have updated the related content in the introduction part of the manuscript. The objective of edge computing is usually to provide computing services to end-users at the edge of the access network. UAV-assisted edge computing is used as a supplementary approach for some special scenarios. It should be noted that most of the application research related to edge computing is still in the theoretical research stage at present, and there is still a long way to go before practical use and commercialization. Related reviews show that edge computing technology will show great potential in video analysis, face recognition, smart city, VR/AR applications, intelligent vehicle applications, and cloud games in the future.
Point 2: Where do you position SACDCO compared to the Fixed Local-UAV Scheme in terms of percentage of allocation of tasks for UAV’s?
Response 2: In our proposal, the offloading rate is defined as the action space of the RL algorithm, which is dynamically varying. Adjusting the offloading rate is part of the offload decision. Offloading rate (as part of the action space) depends highly on the current state space, which includes the current location of the UAV, the UAV's remaining battery capacity, and the size of computing tasks. When the remaining battery capacity of the UAV is high, and the current location of the UAV is close to the end-user, the offloading rate will be set relatively high. Otherwise, it will be set relatively low.
Point 3: In UAV-Only Scheme and Fixed Local-UAV Scheme, having in mind that discarded tasks are correlated with UAV’s battery, why you don't implement a very simple algorithm to reduce distributions of tasks for UAV's that are approaching to the end of battery state? Could you consider such an optimization in order to reduce the discarded tasks values?
Response 3: Thank you for your recommendation. It is noted that the size of discarded tasks is not the only optimization objective in our proposal. Our optimization objective also includes system delay and the energy consumption of the UAV. The reduction of computing tasks allocated by UAV will lead to an increase in system delay, which is contrary to our objective. In Figure 6 of the manuscript, we only compared our proposal with the UAV-only scheme in terms of the size of discarded tasks. And the reason is all the computing tasks of the end-user are offloaded to the UAV, which leads to the exhaustion of the UAV energy, and the remaining tasks are discarded. Compared with other schemes, the energy consumption of the UAV is faster than other schemes, resulting in the size of the discarded tasks being higher than other schemes.

Round 2
Reviewer 3 Report
Congrats for making the requested changes.